# Abscisic Acid—Defensive Player in Flax Response to *Fusarium culmorum* Infection

**DOI:** 10.3390/molecules27092833

**Published:** 2022-04-29

**Authors:** Aleksandra Boba, Kamil Kostyn, Yelyzaveta Kochneva, Wioleta Wojtasik, Justyna Mierziak, Anna Prescha, Beata Augustyniak, Magdalena Grajzer, Jan Szopa, Anna Kulma

**Affiliations:** 1Faculty of Biotechnology, University of Wrocław, Przybyszewskiego 63/77, 51-148 Wroclaw, Poland; yelyzaveta.kochneva@uwr.edu.pl (Y.K.); wioleta.wojtasik@uwr.edu.pl (W.W.); justyna.mierziak@uwr.edu.pl (J.M.); beata.augustyniak@uwr.edu.pl (B.A.); szopa@ibmb.uni.wroc.pl (J.S.); 2Department of Genetics, Plant Breeding & Seed Production, Faculty of Life Sciences and Technology, Wroclaw University of Environmental and Life Sciences, pl. Grunwaldzki 24A, 50-363 Wroclaw, Poland; kamil.kostyn@upwr.edu.pl; 3Department of Food Science and Dietetics, Wroclaw Medical University, Borowska 211, 50-556 Wrocław, Poland; anna.prescha@umw.edu.pl (A.P.); magdalena.grajzer@umw.edu.pl (M.G.)

**Keywords:** flax, *Fusarium culmorum*, infection, abscisic acid, terpenoids

## Abstract

*Fusarium culmorum* is a ubiquitous soil pathogen with a wide host range. In flax (*Linum ussitatissimum*), it causes foot and root rot and accumulation of mycotoxins in flax products. Fungal infections lead to huge losses in the flax industry. Moreover, due to mycotoxin accumulation, flax products constitute a potential threat to the consumers. We discovered that the defense against this pathogen in flax is based on early oxidative burst among others. In flax plants infected with *F. culmorum*, the most affected genes are connected with ROS production and processing, callose synthesis and ABA production. We hypothesize that ABA triggers defense mechanism in flax and is a significant player in a successful response to infection.

## 1. Introduction

Flax (*Linum usitatissimum*) is a plant cultivated all over the world as a source of fiber, oil, and bioactive, health-oriented compounds. In 2020, the production of flax in the European Union exceeded 836 thousand tons, on an area of over 162.5 thousand ha [1]. Unfortunately, flax cultivation is constantly threatened by diseases caused by many different pathogenic microorganisms. Moreover, climate change followed by altered plant cultivation techniques result in the emergence of new and more resistant pathogens which leads to the constant search for new ways to increase the crop plants’ resistance to biogenic threats. The greatest losses in flax cultivation are caused by the specific pathogen of flax, *Fusarium oxysporum* f. sp. *lini*, and non-specific pathogens such as *Fusarium culmorum*, *Alternaria* spp., *Botrytis* spp. [2]. In our previous study, we focused on the flax response to *F. oxysporum* and found that one of the first reactions of the plant to the pathogen is mediated by abscisic acid [3]. In this study, we investigated flax response to *F. culmorum* to find a similar path of response triggered by this pathogen.

*Fusarium culmorum* is a phytopathogenicnon-specific species that causes fusariosis in various parts of plants, most often in roots (FRR—Fusarium root rot) and heads (FHB—Fusarium head blight). It is a cosmopolitan species, most common in regions with a temperate climate [4]. It is a soil-borne pathogen, with a hemibiotrophic life cycle that infects plants by spreading from the remaining plant debris [5]. The fungus produces macrospores (conidia), which is the main factor of spread and infection, and due to the formation of chlamydospores it can survive in the soil for 2–4 years [6]. The pathogen enters the roots of plants 12–24 h after infection and continues to grow [7,8]. The fungus penetrates through the cell walls of the root cells in the tensile zone or at the sites of lateral root formation and grows in epidermal cells, intercellular spaces between the epidermis and cortical layers, and the cortex, spreading upwards to higher parts of the plant. Conidia can also be introduced through the stomata and flowers [8,9]. The sexual stage has not yet been found for this species, and the fungus does not produce microspores [10].

*F. culmorum* produces a number of trichothecene mycotoxins with nivalenol (NIV) and deoxynivalenol (DON) among them. Mycotoxin production is the criterion of their division into two chemotypes: NIV producing and more aggressive DON producing strains [11]. DON participates in pathogenesis by inhibiting the activation of plant resistance genes and in fact, it has been suggested that the production of trichothecenes is the major determinant of the infection spread [7,12].

Pathogenic infection constitutes huge stress in plants, and similar to most stressors, it leads to the production of large amounts of reactive oxygen species in the so-called “oxidative burst”, followed by intensified synthesis of ROS quenchers to prevent cell damage [13]. Accumulation of ROS is closely related to the induced defense response that often allows the expression of a large number of defense related genes, including cell wall proteins, PR proteins, hydrolytic enzymes, enzymes associated with the synthesis of lignin and phytoalexins, transcription factors (TFs), protease inhibitors and signaling compounds (ethylene (ET), jasmonic acid (JA), salicylic acid (SA), abscisic acid (ABA), etc.) [3,14]. Among the transcription factors, those belonging to the WRKY and bHLH families play a major role in PR gene expression and control plant regulatory networks involving hormonal signaling mediators, including ABA in SAR-associated processes. Moreover, several genes controlling antioxidant production (both enzymatic and low molecular mass compounds) are under the control of those TFs. In general, main antioxidant secondary metabolites produced during infection can be divided into two groups: phenylpropanoids and terpenoids. Compounds from the phenylpropanoid pathway such as phenolic acids, flavonoids, anthocyanins participate in plant response to stress, actively by quenching ROS or acting as signaling molecules and passively, by reinforcing the cell wall [15,16]. Terpenoids are the largest group of secondary metabolites and present a gamut of structures, and thus functions. Some may act directly on pathogens, e.g., by disrupting cell membrane, such as thymol or carvacrol [17] by leading to the dysfunction of the fungal mitochondria by inhibiting efflux pumps, or by inhibiting the expression of gene clusters involved in mycotoxin production [18], some are signal or hormonal molecules (ABA, brassinosteroids), but one of their main functions is to act as antioxidants to prevent oxidation of chlorophylls (carotenoids, tocopherols) [19,20].

Induced systemic resistance (ISR) and systemic acquired resistance (SAR) are two forms of induced resistance in plants. They are induced by infection or treatment that results in subsequent resistance against pathogen or parasite. SAR can be triggered by exposing the plant to virulent, avirulent, and nonpathogenic microbes and is connected with SA production and signaling [21]. In SAR, activation of defense genes including those encoding for PRs takes place before the upper leaf is reached by a pathogen. Increased flavonoid and anthocyanin synthesis occurs (activation of genes encoding chalcone synthase, chalcone isomerase, dihydroflavonol reductase, leucoanthocyanidin dioxygenase, numerous glucosyltransferases and others). No salicylic acid nor PR proteins participate in ISR, ethylene and jasmonate signaling is observed, though not always required, and priming rather than direct activation of defense is predominantly observed [22]. To some extent both plant responses are similar. As ROS act as a long-distance messenger in plants, genes involved in their synthesis and processing are activated in plants during resistance response [23]. *RbohD* (NADPH/respiratory burst oxidase homolog protein D) gene activation encoding NADPH oxidase (*NOX*) was shown to be required for the initiation and self-propagation of a rapid cell to cell systemic signal of H_2_O_2_ spreading through the extracellular spaces [24]. To keep the spread of free radicals in check, activation of low molecular weight antioxidants (such as phenylpropanoids) biosynthesis genes (phenylalanine ammonia lyase (*PAL*)) and genes encoding antioxidative enzymes (e.g., peroxidase gene, catalase gene) must be ensured. However, especially in the early stages, it is difficult to distinguish between SAR and ISR, both of which activate genes connected with the response to pathogen invasion and increase the accumulation of reactive oxygen species (ROS) and callose. It is known that different pathogenic species belonging to the same genus may trigger different responses in the target plant. Fungi of *Fusarium* genus led to the development of ISR in early stages of infection, and later on, SAR is activated [25,26,27]. It is known that *F. oxysporum* causes increased production of salicylic acid and methyl-salicylates, which indicates SAR activation in the infected flax. However, no information on *F. culmorum* infection mechanism in flax is available. Based on our previous study we also established that ABA plays a crucial role in the early moments of infection of flax with *F. oxysporum*.

Although the role of ABA in biogenic stress response is still ambiguous there is more and more information on its involvement in plants’ response to pathogen infection [28,29]. Plant resistance to pathogens was reported to be negatively affected by ABA. For instance, ABA suppressed immune responses of tomato infected with *Botrytis cinerea* [30] barley infected by *Magnaporthe oryzae* [31], or tobacco infected by *Ralstonia solanacearum* [28]. On the other hand, ABA increased resistance of *Arabidopsis* to *Alternaria brassicicola* [32] and was involved in higher resistance of flax to *Fusarium oxysporum* [3]. The contrasting roles of ABA in plant defenses seem to be depending on the infection phase and pathogen lifestyle. Bearing this in mind studying plant responses to infection should be exerted for individual species and infection phases. It has been proposed that in general, the way ABA acts in response to infection depends on the pathogen type. It is long known that ABA is responsible for stomatal closure in the earliest phase of infection, both bacterial and fungal. In the subsequent phases the role of ABA varies, for instance, in contrast to fungal infection, a bacterial attack would not lead to callose synthesis activation by ABA [33]. There is also evidence that ABA influences resistance against herbivorous insects. For example, ABA-deficient tomato (*Solanum lycopersicum*) and *Arabidopsis thaliana* mutants have been reported to be more susceptible to infestation by insects [34]. To broaden the pool of data indicating the above, in this work we investigated a possible effect of *F. culmorum* on ABA with the consequences of this hormone action. However, the mechanisms of infection with *Fusarium* in flax are unknown. In this study, we intended to shed light on some aspects of the irrefutably complex mechanism of plants’ response to *F. culmorum*, especially regarding the role of the terpenoid pathway and ABA.

## 2. Materials and Methods

### 2.1. Biological Material and Conditions of Flax Seedling Growth

Flax seeds (*Linum usitatissimum*, var. Nike) were obtained from the Flax and Hemp Collection of the Institute of Natural Fibers and Medicinal Plants in Poland. The pathogenic strain of *Fusarium culmorum* was obtained from the Plant Breeding and Acclimatization Institute in Poznan, Poland. The flax seeds were sterilized in a 50% PPM solution (Plant Cell Technology, Washington, DC, USA) for 10 min, rinsed with sterile water, and then placed in Petri dishes with solidified 0.9% agar Murashige and Skoog (MS) medium. Next, the dishes were placed for 14 days in a phytotron, in a 16 h light (21 °C), 8 h darkness (16 °C) regime. Using *F. culmorum* cultures prepared earlier (grown on PDA medium) equal aliquots of mycelium were inoculated into the Petri dishes containing 20 mL of PDA medium. Cultures were incubated for 7 days, then the 14-day-old flax seedlings were transferred together with the MS medium onto the dishes with the fungi (the plants’ roots were in direct contact with the mycelium, Appendix A). After the appropriate time of induction (6, 12, 24, 36 or 48 h), the plants were collected, frozen in liquid nitrogen, and stored at −80 °C for further analyses.

### 2.2. H_2_O_2_ Content Determination in Flax Seedlings after Infection

Plant material was ground in liquid nitrogen using a mortar and pestle and 50 mg of the obtained powder was used for processing. The ground tissue was suspended in 200 µL of 20 mM phosphate buffer pH 6.5 and centrifuged (10 min, 4 °C, 10,000× *g*). The supernatant was transferred to a new Eppendorf tube and 50 µL were used for H_2_O_2_ content determination. The assay was performed using Amplex™ Red Hydrogen kit (Life Technologies, Carlsbad, CA, USA) according to the producer’s instructions. Varioskan Flash (Thermo Scientific, Boston, MA, USA) was used to obtain the fluorescence of the probes.

### 2.3. Reverse Transcription Real-Time PCR

Total RNA was isolated with Trizol according to the manufacturer’s protocol (Life Technologies). Purification of the samples from genomic DNA was performed with DNase I as described in the manufacturer’s protocol (Thermo Scientific). Isolated plant’s RNA was transcribed to cDNA using a High-Capacity cDNA Reverse Transcription Kit (Applied Biosystems, San Francisco, CA, USA). Real-time RT-PCR was performed on the Applied Biosystems StepOnePlus Real-Time PCR System. The DyNAmo HS SYBR Green qPCR Kit (Thermo Scientific) was used for reaction. The list of primers used in the real-time RT-PCR is shown in Appendix A. The actin gene was used as a reference gene. The next parameters were selected for the reaction: 95 °C for 15 min (holding stage); and 95 °C for 10 s, 57 °C for 20 s, 72 °C for 30 s, 37 cycles (cycling stage). The conditions of the melting curve stage were: 95 °C for 15 s, 60 °C for 1 min, 95 °C for 30 s, and ramp rate: 1.5%.

### 2.4. Isolation and Analysis of Terpenoids

For the terpenoids isolation 25 mg of lyophilized seedling tissue was used. The extraction had three steps: the terpenoid compounds were sequentially extracted with acetone, petroleum ether and diethyl ether for 10 min in an ultrasonic bath, and centrifuged at 10,000× *g* and 4 °C for 10 min. The supernatants were collected, combined and dried under nitrogen and resuspended in 0.5 mL of methanol/acetonitrile mixture. The extraction was performed in three repetitions. The extraction method has been previously described [35]. The samples were then analyzed in UPLC with a 2996 PDA diode detector (Waters Acquity UPLC System) on a BEH C18 column, 2.1 × 100 mm, 1.7 μm (Waters Acquity). Eluent A was a mixture of 50% methanol, 40% acetonitrile and 10% H_2_O and eluent B was 50% methanol and 50% acetonitrile. The separation of the analyzed compounds was conducted at a 0.5 mL/min flow rate with the following program: 0–1 min 100% A, 2–9 min 100% B, 10 min 100% A. Comparison to authentic standards (Sigma-Aldrich, St. Louis, MO, USA) was used for components identification and their quantification. The integration of peaks for carotenoids was conducted at 475 nm and for tocopherols at 290 nm.

### 2.5. Isolation and Analysis of Sterols

The plant tissue for the analysis was ground in liquid nitrogen and then lyophilized. Three ml of methanol:chloroform (2:1, *v*/*v*) solution was added to 200 mg of plant tissue and incubated at room temperature for 30 min. Then, the samples were supplemented with 1 mL of chloroform and 1.8 mL of H_2_O, mixed and centrifuged. The chloroform layer was collected in a new tube and dried under nitrogen flow. The resulting pellet was dissolved in 2 mL of 2 M KOH in a glass tube with a PTFE (polytetrafluoroethylene) cap. 30 μL of 5α-cholestane solved in hexane has been added to the samples as an internal standard. The samples were mixed and incubated at 70 °C for 45 min and cooled down afterward. In the next step 1 mL of H_2_O, 0.5 mL of 96% ethanol and 1.5 mL of hexane were added. The samples were shaken for 5 min (Multi Reax Shaker) and centrifuged for 5 min at 500× *g*. After that, the hexane layer was collected in new tubes and the procedure was repeated twice. The hexane fractions were combined, dried under nitrogen flow and silylated (BSTFA in 1% TMCS). The samples were shaken for 2 min, incubated at 70 °C for 45 min and were dried under nitrogen flow and resuspended in 200 μL of hexane. Sterol analysis was performed with a gas chromatograph (Agilent Technologies 7890A, Santa Clara, CA, USA) with a flame ionization detector (FID). Qualitative and quantitative analysis was conducted based on comparing the obtained chromatograms with a library of chromatograms of pure standards.

### 2.6. Abscisic Acid Isolation and Quantification

Plant tissues were ground in liquid nitrogen directly before ABA isolation. For 200 mg of plant tissue, 500 μL of extraction buffer (90% methanol, 200 μg/mL dithiocarb) was added. The samples were mixed and transferred to glass tubes and incubated at 4 °C overnight. After isolation, the samples were centrifuged for 10 min at 6000× *g*, 4 °C. The supernatant was transferred to a microtube and dried in a vacuum drier at 4 °C, and the pellet was then resuspended in 400 μL of buffer containing 10% methanol, 50 mM Tris pH 8.0, 1 mM MgCl_2_, and 150 mM NaCl. For quantitative analysis of the ABA Phytodetek Immunoassay Kit for ABA (Agdia Inc., Elkhart, IN, USA) kit was used.

### 2.7. Callose Isolation and Quantification

For the callose isolation, 50 mg of lyophilized plant tissue was washed once with 96% ethanol and three times with 20% ethanol. Then, 1 mL of 1 M NaOH was added to the sample. The samples were incubated for 15 min at 80 °C and centrifuged for 15 min, 8000× *g*. Then, 40 μL of 0.1% (*w*/*v*) aniline blue, 21 μL of 1 M HCl and 59 μL of 1 M glycine–NaOH (pH 9.5) buffer were added to 20 μL of the collected supernatant, mixed and incubated for 20 min at 50 °C and for 30 min at room temperature. Callose content was measured at the spectrofluorometer at 393 nm excitation wavelength and 484 nm emission wavelength. Curdlan (β-1,3-glucane) was used for standard curve preparation.

### 2.8. Flax Seedling Treatment with Abscisic Acid

Four-week-old flax seedlings were sprayed with ABA water solutions (in six different concentrations: 2.5 µM, 5 µM, 10 µM, 25 µM, 50 µM, 100 µM). Pure water spraying was used as a control. The tissues were collected in 24 h after the treatments, frozen in liquid nitrogen and stored at −80 °C before analyses.

### 2.9. Treatment of F. culmorum with ABA

The impact of abscisic acid on *F. culmorum* growth was determined by supplementing PDA media with ABA in five concentrations: 0.1 μM, 0.5 μM, 1 μM, 10 μM and 50 μM. *F. culmorum* (the equal aliquots of mycelium) were grown on PDA for 14 days at 28 °C in darkness and checked daily for changes. Petri dishes with PDA medium and the appropriate amounts of solvent were used as a control.

### 2.10. Treatment of F. culmorum with Carotenoids and Tocopherol

To evaluate the influence of carotenoids and tocopherols on *F. culmorum* mycelium, growth mycelium fragments were placed on PDA media supplemented with authentic standards (Sigma-Aldrich) of lutein, β-carotene, α-tocopherol and γ-tocopherol at 0.5 μM concentrations. After 3–5 days of growth at 28 °C, the diameters of the mycelia were measured and photographed. Pure solvent, in which the standards were dissolved, was used as the control.

### 2.11. Statistical Analysis

The experiments were performed in three biological repetitions. We used Tukey and Kruskal–Wallis tests to evaluate statistical significance. The differences were statistically significant at *p* < 0.05.

## 3. Results

### 3.1. Determination of H_2_O_2_ Level in Flax Seedlings after F. culmorum Infection

One of the first reactions of the plant to pathogen infection is the production of reactive oxygen species (ROS) in the process of oxidative burst. Among the produced ROS molecules, measuring H_2_O_2_ concentration seems to be a good indicator of the plant’s reaction, as the molecule is sufficiently stable to be measured precisely. To track the response of infected flax seedlings we measured the H_2_O_2_ levels at different time points after the inoculation with *F. culmorum*. The amount of H_2_O_2_ after the infection was observed to reach the highest level measured at 36 hpi (hours post-infection) (530 nmols per 1 g of fresh weight) and dropped back to the control level at the last time-point (48 hpi) (Figure 1).

### 3.2. Analysis of Transcript Levels of Genes Involved in ROS Processing

We measured the level of transcripts of genes encoding enzymes involved in the processing of ROS in the infected flax seedlings, namely NADPH oxidase (*RboH*) ascorbate peroxidase (*APx*), catalase (*CAT*), and three superoxide dismutases (*SOD-Fe*, *SOD-Mn*, *SOD-Cu*). The transcript level of NADPH oxidases increased significantly. Especially, the level of *RboHD* gene transcription was increased to 8.8-, 12.2- and 19.4-fold of the control at 6 hpi, 12-hpi and 24 hpi, respectively, and to over 50-fold of the control at 36 hpi and 48 hpi. The levels of *APx* gene transcript were elevated at 24 hpi, 36 hpi, and 48 hpi, and the increase was ca. 2-, 1.5- and 2-fold of the control. A similar tendency was observed for *CAT* gene, but its transcript level was decreased in the first two time-points to around 0.5-fold of the control. While the iron dismutase gene seemed to be unchanged in all measurements (except the earliest time-point, where it was down to 0.5-fold of the control), the two remaining ones were activated, though the activation was low in the case of *SOD-Mn*. The activation of *SOD-Cu* was observed for all time-points starting with 12 hpi (Figure 2).

### 3.3. Analysis of Transcript Levels of Genes Involved in Isoprenoid Biosynthesis

Two general routes are involved in the biosynthesis of isoprenoids in plants. The mevalonate pathway (MP) is carried out in the cytosol, and the non-mevalonate pathway (NMP) in plastids. Generally, the MP is responsible for phytosterol biosynthesis, whereas the NMP is greatly branched giving rise to a plethora of different compounds, with chlorophylls, tocopherols, carotenoids, and some phytohormones (GA, ABA) among them. Both of the pathways start with the synthesis of two isomers, isopentenyl pyrophosphate (IPP), and dimethylallyl pyrophosphate (DMAPP), though different enzymatic machineries are utilized. We measured the transcript levels of the genes (called here the terpenoid backbone synthesis genes) involved in the biosynthesis of those isomers (Figure 3A). In contrast to deoxy-d-xylulose 5-phosphate synthase (*DXS*) gene expression, which was decreased in all time-points measured (even to 0.57-fold and 0.51-fold of the control at 24 hpi and 36 hpi, respectively), the transcript level of deoxy-d-xylulose 5-phosphate reductase (*DXR*) gene was increased in all time-points after infection. The activation of this gene reached 2.67- and 2.5-fold of the control at 12 hpi and 24 hpi, respectively, and maximally 3.23-fold of the control at 48 hpi. Similarly, the expression levels of isopentenyl pyrophosphate isomerase (*IDI*) and geranyl pyrophosphate synthase (*GPPS*) genes were increased. The activation of *IDI* gene increased in each time-point, whereas the maximum expression (ca. 2.5-fold of the control) of *GPPS* gene was measured at 24 hpi and 36 hpi. In the cytoplasmic route the expressions of 3-hydroxy-3-methylglutaryl-CoA synthase and reductase genes (*HMGS* and *HMGR*, respectively) were not changed significantly relative to the control (0.75-fold of the control for *HMGS* at 36 hpi). The expression of mevalonate kinase (*MVK*) was increased to 1.75-fold of the control at 36 hpi. The transcript level of farnesyl pyrophosphate synthase (*FPPS*) was decreased in all time-points with minimum expression at 0.23-fold of the control at 36 hpi.

Genes of the cytoplasmic route of isoprenoid biosynthesis play a role in the production of phytosterols and brassinosteroids. After measuring the transcript levels of genes involved in these compounds’ biosynthesis, we observed a general inactivation of this pathway (Figure 3B). The minimal expression was observed at 24 hpi and 36 hpi (except sterol methyltransferase 2 (*SMT2*) gene, for which the expression was 1.5-fold of the control at 24 hpi). In contrast, activation of *CYP710A* gene encoding a cytochrome p450 protein responsible for desaturation of the double bond at the C-22 position in the side chain was observed in all time-points, with a maximum (9.1-fold of the control) at 48 hpi.

Measuring the expression of genes involved in tocopherol biosynthesis we noted a consequent activation of this branch of the pathway except geranyl-geranyl diphosphate reductase (*GGR*) gene, which, excluding the 6 hpi, was decreased in all remaining time-points (to 0.17-fold of the control at 36 hpi). Although the transcript levels of homogentisate phytyl transferase (*VTE2*), MPBQ methyltransferase (*VTE3*), tocopherol cyclase (*VTE1*) and γ-tocopherol methyltransferase (*VTE4*) were virtually all increased at an early time- points at 36 hpi all of them showed a sharp dropdown with minimal value of 0.26-fold of the control for *VTE3* (Figure 3C).

We analyzed twelve genes involved in the biosynthesis of carotenoids: phytoene synthase (*PSY*), phytoene desaturase (*PDS*), ζ-carotene desaturase (*ZCD*), ζ-carotene isomerase (*Z-ISO*), carotenoid isomerase (*CRTISO*), lycopene β-cyclase (*LCYB*), lycopene ε-cyclase (*LCYE*), β-carotene hydroxylase (*βHY*), ε-ring hydroxylase (*LUT1*), β-ring hydroxylase (*LUT5*), zeaxanthin epoxidase (*ZEP*), violaxanthin de-epoxidase (*VDE*) at 6, 12, 24, 36 and 48 hpi. We observed an increase in the PSY gene transcript to 2.7- and 3.6-fold of the control at 24 hpi and 48 hpi, respectively. Although small in the beginning, an increase in the expression of Z-ISO gene was observed in each consecutive time-point reaching 3-fold of the control at 48 hpi. A similar expression profile was noted for CRTISO gene (with a maximum of 2.37- and 2.71-fold of the control at 36 hpi and 48 hpi, respectively. ZDS gene transcript level was increased at 12 hpi and 48 hpi to 1.55- and 1.76-fold of the control, respectively. Expressions of both lycopene cyclase genes (LCYB, LCYE) were decreased after the infection (only with 1.22-fold increase at 48 hpi, and 1.46-fold increase at 6 hpi, respectively). However, a considerable increase in the expression of βHY gene throughout all time-points was measured, reaching 3.65- and 3.88-fold of the control at 36 hpi and 48 hpi, respectively. Both ZEP and VDE gene expressions were slightly decreased in all time-points measured. LUT5 gene transcript level decreased till 36 hpi to increase at 48 hpi to 1.38-fold of the control, whereas the LUT1 gene expression changed over the time-points to reach its minimum at 36 hpi (0.46-fold of the control) and maximum at 48 hpi (1.46-fold of the control) (Figure 3D).

Since further processing of carotenoids, connected with their degradation, leads to the production of other relevant compounds, such as strigolactones, volatile apocarotenoids and perhaps the most important to ABA, we analyzed the levels of expression of genes involved in this process: 9-cis-epoxycarotenoid dioxygenases (*NCED3*, *NCED6*), xanthoxin dehydrogenase (*ABA2*), abscisic aldehyde oxidase (*AAO3*), carotenoid cleavage dioxygenases (*CCD1*, *CCD7*, *CCD8*). We observed a substantial increase in the level of the *NCED3* and *NCED6* gene levels (up to 10.89- and 22.66-fold of the control, respectively at 36 hpi). Additionally, the level of *AAO3* gene expression was elevated with its maximum of 3.79-fold of the control at 36 hpi, whereas the expression of *ABA2* gene was decreased in all time-points (minimum of 0.63-fold of the control at 36 hpi). Expression levels of all three *CCD* genes were down. In the case of *CCD1*, the decrease was relatively small, whereas for *CCD7* and *CCD8* the dropdown was considerable through all the time-points (to 0.28- and 0.53-fold of the control on average, respectively) (Figure 3E).

### 3.4. Analysis of Selected Isoprenoid Levels

We measured the amounts of selected isoprenoids in the flax tissue treated with *F. culmorum* at 6 hpi, 12 hpi, 24 hpi, 36 hpi, and 48 hpi. The total level of phytosterols was somewhat (ca. 5%) elevated in the measured time-points (except the 6 hpi, where it was 78.4% of the control). Particular phytosterols were quantified and are presented in Figure 4A. In general, decreased levels of all measured phytosterols were observed at 6 hpi and slight, but consistent increase at 36 hpi and 48 hpi.

Similar as it was in the case of sterols, the levels of all measured carotenoids, as well as chlorophylls were down to 85% of the control on average at the beginning of the infection, but then their levels increased to reach maximum 1.24-fold of the control for violaxanthin at 12 hpi, 1.11-fold of the control for tocopherols at 24 hpi to drop down to the control level at 36 hpi. Then, at 48 hpi, a decrease was noted for all of the compounds in this group, by 12% compared with the control on average (Figure 4B). The level of ABA was considerably increased throughout all the time-points investigated and reached maximally 7.6-fold of the control at 12 hpi (Figure 4C).

### 3.5. Analysis of the Expression Level of Callose Synthase Gene and Callose Content in Response to F. culmorum Infection

ABA is considered a regulator in plants’ response to infection and influences many of defense genes that are involved in the biosynthesis of compounds participating in the resistance to pathogens, such as callose. We measured the transcript levels of callose synthase genes (Figure 5). Although inconsistent, the transcription of *CALS1* (callose synthase 1) gene was elevated after the infection reaching its maximum at 48 hpi (2.03-fold of the control). Similarly, the level of *CALS3* (callose synthase 3) gene was increased in the measured time-points and the increase was most prominent at 36 hpi and 48 hpi (2.17- and 2.07-fold of the control, respectively). *CALS2* (callose synthase 2) gene transcript level was virtually at the control level in all time-points with somewhat increase at 48 hpi (1.39-fold of the control). *CALS4* (callose synthase 4) gene expression decreased at 24 hpi and 36 hpi (0.68- and 0.31-fold of the control, respectively) to return to the control level at 48 hpi.

We measured the amounts of callose at 6 hpi, 12 hpi, 24 hpi, 36 hpi, and 48 hpi. After a decrease observed in 24 hpi to 0.69-fold of the control, its level started to grow up to reach 2.15-fold of the control at 48 hpi (Figure 6).

### 3.6. Influence of Carotenoids on F. culmorum Growth

We investigated the direct effect of carotenoids and tocopherol on *F. culmorum* growth, yet we did not observe any changes. The areas of the fungal mycelia were comparable with the control (Appendix A), though we observed changes in colorization of the mycelium in some cases, which suggest that some of the metabolites are actively metabolized by the fungus.

### 3.7. Influence of ABA on F. culmorum Growth

We grew *Fusarium culmorum* on PDA medium supplemented with different concentrations of ABA (0.1 μM, 0.5 μM, 1 μM, 10 μM and 50 μM) for 14 days. We did not observe any changes in the fungus appearance nor growth compared with the control (data not shown).

## 4. Discussion

There are numerous pieces of data in the literature on the participation of PR proteins, salicylic acid and jasmonic acid, as well as antioxidative compounds in plant resistance to fungal infection [36,37,38]. The role of plant hormones in responses to environmental stresses has been proven, and the activation of the synthesis of these compounds has been well researched [39,40]. However, some of them are better researched than others and in some cases such as ABA their role in abiotic stress is much better understood than biotic interaction. For years abiotic and biotic stress signaling were usually researched separately even though plants have to deal simultaneously with several stresses of differing nature at once [41]. It is clear that we are still far from the full picture of the mechanisms involved in plant response to stress conditions. In particular, the role of abscisic acid in plant–pathogen interactions remains unclear with contradictory evidence presented in the literature. However, recently ABA is being recognized as an important molecule modulating immune response [42]. Previously, the majority of work was carried out in the *Arabidopsis* model system, and it becomes evident that other species may respond differently and plant–pathogen interactions need to be studied on case-by-case basis [43].

In our former study, we evidenced ABA involvement in flax resistance to *Fusarium oxysporum* at early stages of infection [3]. It was manifested among others by elevated levels of *PMR4* gene expression accompanied with higher levels of callose synthesized at 36 hpi and 48 hpi. In this study we show that the response of flax to *Fusarium culmorum* infection exhibits some similarities with those studied previously. Most importantly, we observed increase in ABA levels in flax seedlings infected with *F. culmorum* correlating well with transcript levels of ABA synthesis genes. It is estimated that at least 10% of protein-coding genes have ABA-responsive elements in their promoters [44], and a lot of induced genes overlap with those involved in pathogen response [45]. We also observed a similar profile of the expression of the terpenoid biosynthesis genes compared with that noted for *F. oxysporum* infection. Activation of the core genes (*DXR*, *IDI*, *GPPS*, *HMGR*, *MVK*), mutual for both the cytoplasmic and plastidial parts of the pathway was observed, which was consistent with the data obtained for *F. oxysporum*. Similarly, phytosterol synthesis was inhibited, which translated to the activation of the plastidial branch of this pathway, from the earliest time-point studied (6 hpi). It may be connected to the need to protect the photosystems from free radicals produced after the first contact of the plant with the pathogen in so- called oxidative burst. In contrast to *F. oxysporum* infection, where there were practically no changes in the terpenoid levels, we measured some increase in the carotenoids, especially at 12 hpi, but also at 24 hpi and 36 hpi. No direct correlation with the gene transcript levels was found. However, it may be due to the fact that activation of the key genes of this branch (*PSY*, *PDS*, *Z-ISO*, *ZCD* and *CRTISO*) is critical, whereas others (*LCYB*, *LCYE*) may be inhibited as the pool of corresponding enzymes is sufficient to produce carotenoids [46].

Complex organisms, including plants, react to microorganism infection with rapid production of reactive oxygen species (ROS). ROS play a central role in the defense process of plants. The major source of cellular ROS are NADPH oxidases (NOX), which use NADPH as an electron donor in O_2_ molecule reduction. NOXs were shown to play an essential role in plants’ immunity. For instance, *Arabidopsis* mutants with impaired NOX function (*AtrbohD*, *AtrbohF*) were more susceptible to pathogens compared with the wild type [47]. It was evidenced that respiratory burst oxidase homolog protein D (GhRbohD) positively regulates cotton resistance to *Verticillium dahlia* [48]. Numerous studies provided evidence that the NOXs’ role in plant immunity comes from their interactions with other immunity signal transducers such as Rac/Rop small GTPases [49] or ABA [50]. Thus, ABA may be perceived as ROS inducer, and in fact such observations were made for abiotically stressed plants where it is connected with stomatal closure [51] and during pathogen infection [52]. Our results show increased levels of NOX gene expression (*RbohD*, *RbohF*). Moreover, a positive correlation (0.79) between *RbohF* and ABA level can be noted. In addition, we noticed an increase in *RbohD* and *RbohF* transcript levels in flax treated with (+)ABA (in different concentrations ranging from 2.5 μM to 1 mM). No H_2_O_2_ level change was detected though, which may be due to the activation of genes encoding the enzymes processing the reactive oxygen forms (catalase and peroxidase) (Appendix A). A similar observation of H_2_O_2_ level and those enzymes was made for the *F. culmorum* treated flax. Increased activity of ascorbate peroxidase (*APx*) gene (growing in each subsequent time-point) was determined. Additionally, transcript level of catalase was increased, though later compared with that of *APx*. Peroxidases are involved in a broad range of physiological processes throughout the plant life cycle and are important players in plant defense reaction. Among others, they are involved in lignin and suberin formation, cross-linking of cell wall components, and the metabolism of ROS and RNS (reactive nitrogen species). Moreover, peroxidases may play a role in the ABA-dependent signaling pathway, activated both under abiotic and biotic stress conditions [53].

Although the H_2_O_2_ assay showed no statistically significant differences between *F. culmorum* infected and untreated control flax seedlings it is known that not only hydrogen peroxide is involved in signaling but also NO and Ca^2+^ can take part in plant responses to stresses [54]. Especially Ca^2+^ was recently discovered as an initial factor in plant adaptation to difficult environmental conditions [55]. Worth noting is the existence of a cross talk between free radicals and this should be investigated in flax response to Fc infection in a wider perspective because NO may cause inhibition of hydrogen peroxide accumulation as it was proven for lettuce [56].

Changes detected in the transcript levels of callose synthesis genes correspond with elevated amounts of callose in 48 hpi. Callose deposits together with other possible reinforcement of cell wall through peroxidase activity may greatly contribute to the formation of physical barrier, which facilitates stopping further infection as well as spreading it to the neighboring tissues [57]. Induced callose deposition triggered by holaphyllamine (steroid alkaloid) has been previously detected in flax after inoculation with *F. oxysporum* [58]. Furthermore, ABA-dependent triggering of callose synthesis and deposition was proved as a mechanism in necrotrophic pathogens within *β*-amino-butyric acid-induced response [33]. It can be assumed that in flax containing increased level of ABA the amount of callose also rises. Based on literature evidence, we believe that it is an effect directly corresponding with the infection [3,59,60]. Alternative way, presented for *Arabidopsis thaliana,* involves decreased level of callose synthase leading to complete loss of callose and initiation of disease resistance based on salicylic acid [61].

This work is the first to show the possible mechanism of the effect of hemibiotrophic *Fusarium culmorum* on the immune response in flax driven via ROS production and ABA biosynthesis. As far as we know, the terpenoid pathway in response to this pathogen was studied in detail for the first time. Moreover, it presents gene expression pattern through the early stages of infection, which can be meaningful to understanding strategies implied by flax to protect themselves against pathogenic fungi.

## Figures and Tables

**Figure 1 molecules-27-02833-f001:**
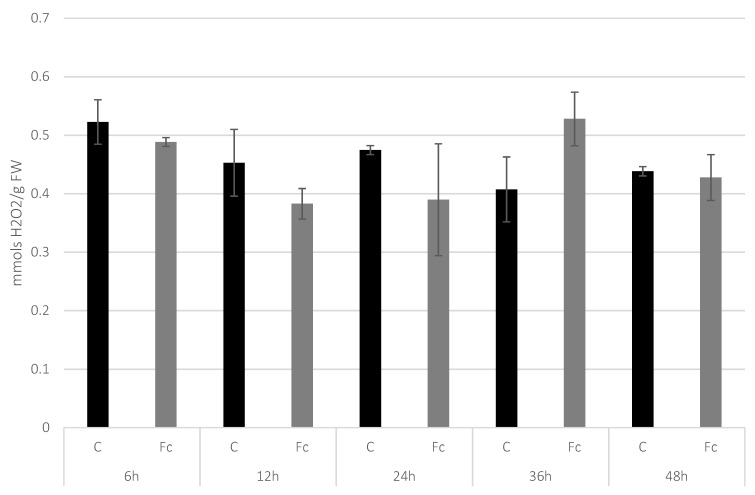
Amounts of H_2_O_2_ measured in 14 days old flax seedlings infected with *F. culmorum* at 6 hpi, 12 hpi, 24 hpi, 36 hpi and 48 hpi. C—untreated control, Fc—*Fusarium culmorum* treated. Results are presented as means of three biological repeats ± sd.

**Figure 2 molecules-27-02833-f002:**
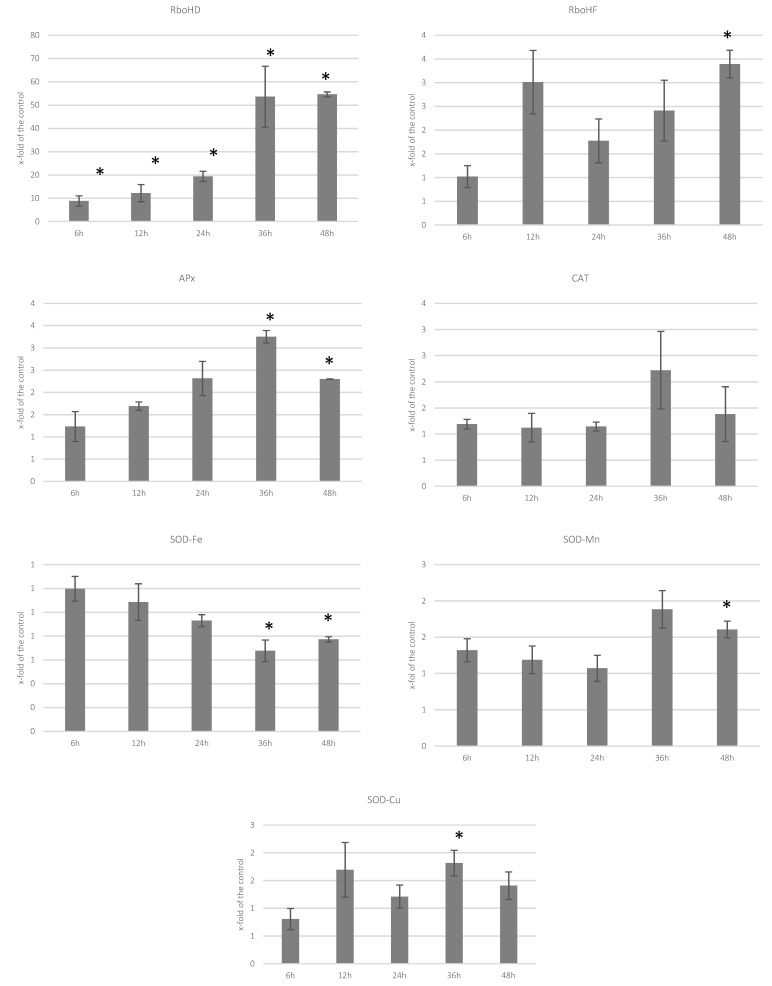
Transcript levels of genes involved in ROS production and processing (*RboHD*—NADPH oxidase D, *RboHF*—NADPH oxidase F, *APx*—ascorbate peroxidase, *CAT*—catalase, *SOD*—superoxide dismutase) in flax seedlings treated with *F. culmorum* measured at five time-points after the infection. Results (means of three biological repeats ± sd) are presented as fold of the control (equal to 1). Statistically significant changes are marked with asterisks.

**Figure 3 molecules-27-02833-f003:**
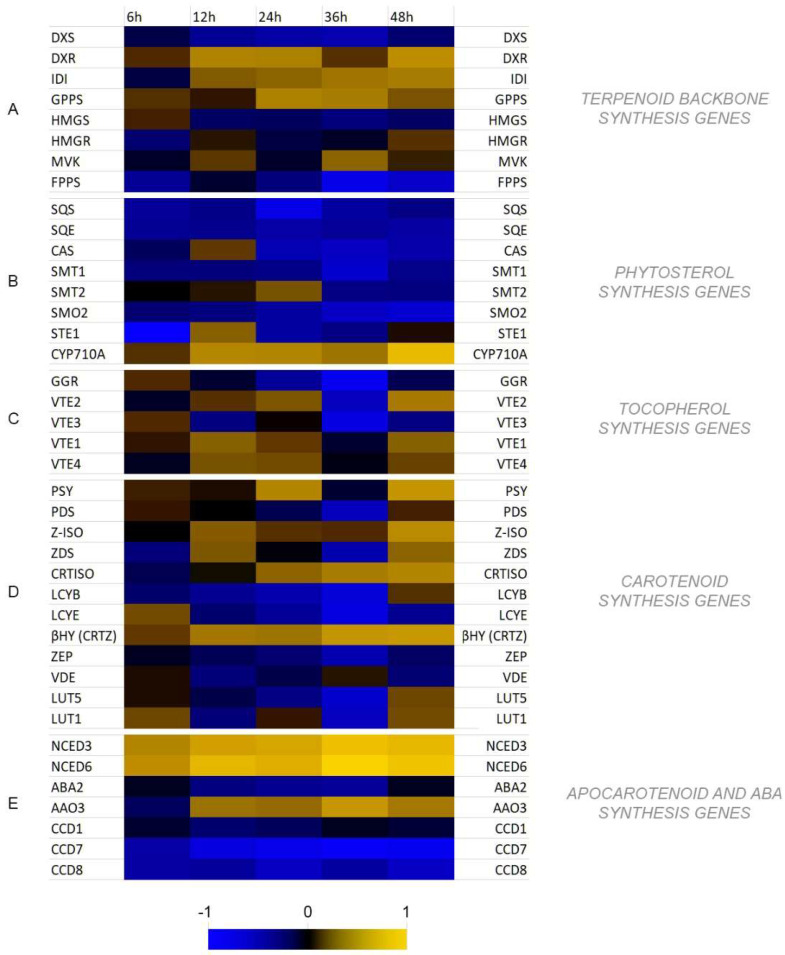
Heatmap of terpenoid pathway key gene expression in flax after *F. culmorum* infection (x—fold of the control). (**A**)—terpenoid backbone synthesis genes; (**B**)—phytosterol synthesis genes; (**C**)—tocopherol synthesis genes; (**D**)—carotenoid synthesis genes; (**E**)—apocarotenoid and ABA synthesis genes. Detailed results on the expression can be found in Supplementary File S3.

**Figure 4 molecules-27-02833-f004:**
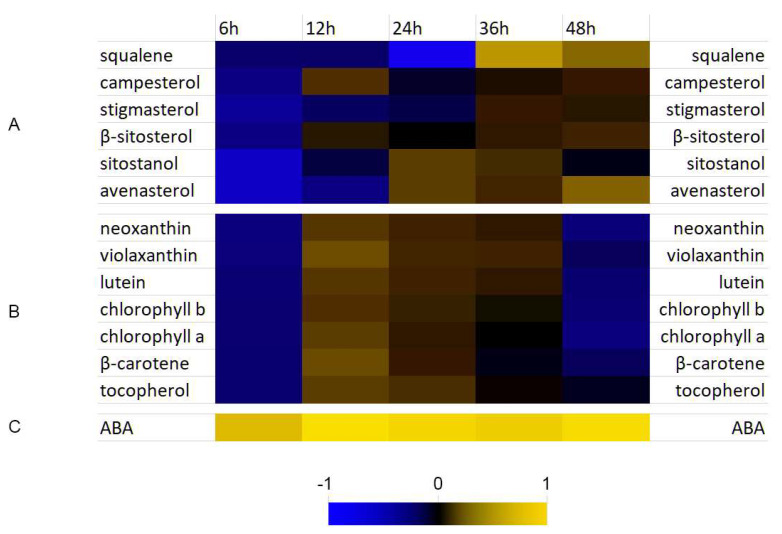
Heatmap of terpenoid metabolites in flax after *F. culmorum* infection (x–fold of the control). (**A**)—phytosterols; (**B**)—carotenoids; (**C**)—ABA. Detailed results on the expression can be found in Supplementary File S4.

**Figure 5 molecules-27-02833-f005:**
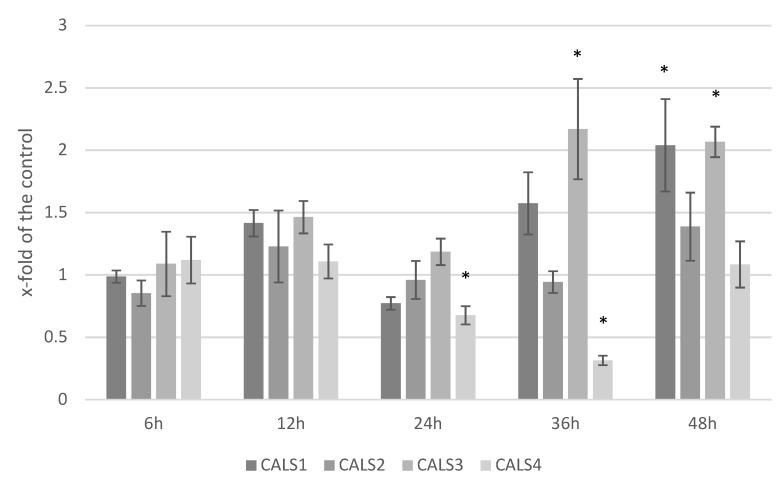
Transcript levels of callose synthase genes (*CALS1*, *CALS2*, *CALS3*, *CALS4*) in flax seedlings treated with *F. culmorum* measured at five timepoints after the infection. Results are presented as a fold of the control (means of three biological repeats ± sd). Statistically significant changes are marked with asterisks.

**Figure 6 molecules-27-02833-f006:**
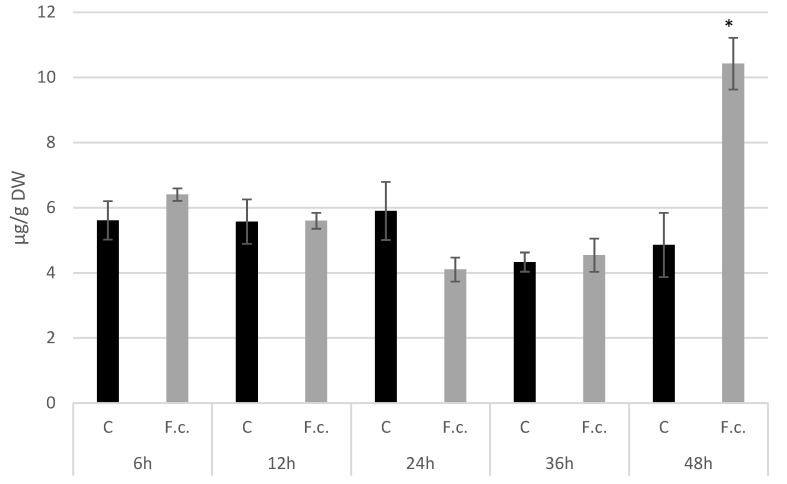
Amounts of callose measured in 14 days old flax seedlings infected with *F. culmorum* at 6 hpi, 12 hpi, 24 hpi, 36 hpi and 48 hpi. C—untreated control, Fc—*Fusarium culmorum* treated. Results are presented as means of three biological repeats ± sd. Statistically significant changes are marked with asterisks.

## Data Availability

Data available on request.

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
