# Peer review of "Abscisic Acid—Defensive Player in Flax Response to Fusarium culmorum Infection"

_molecules, 2022, doi:10.3390/molecules27092833_

Round 1
Reviewer 1 Report
The Materials and Methods section is not clear (especially 2.1. Biological material and conditions of flax seedling growth section).
Two days of fungal growth may not be enough time to induce pathogenesis and related responses in Petri dishes. Ideally, the fungus should cover the whole Petri dish (probably 7-10 days) and flax seedlings can be placed onto those fungal cultures. Then meaningful results can be achieved.
Reviewer 2 Report
Fusarium culmorum can host flax, which causes root rot and mycotoxin accumulation in flax products.Fungal infections can cause huge losses in the flax industry.Potential threat to consumers.The authors found that defense against Fusarium yellow is based on early oxidative bursts in flax.In flax plants infected with Fusarium yellow, the most affected genes are related to ROS production and processing, callose synthesis, and ABA production.Studies have shown that the increase of ABA biosynthesis is caused by the activation of terpenoid biosynthesis pathway.This work demonstrated for the first time the effects of fusarium infection on the immune response of flax, and provided new experimental results and ideas for subsequent experimental studies.However, there are still some detail errors that need to be fixed in this article.
1. In line 371, the authors state that there is a lot of data on the involvement of PR proteins, salicylic and jasmonic acids, and antioxidant compounds in plant anti-fungal infections.Is there any evidence for their role in plant responses to environmental stress?
2. In line 439, the authors ask, "Based on the literature, what evidence can be found for an increase in corpus callosum in flax containing increased ABA levels as an effect directly related to infection?"
3. On line 439, the authors ask, "Based on the literature, what evidence can be found for an increase in corpus callosum in flax containing increased ABA levels as an effect directly related to infection?"Please attach relevant references or cited evidence
4. Lines 86~92 are obviously inconsistent and need to be modified
5. Lines 399~419 are obviously inconsistent and need to be modified
Reviewer 3 Report
Dear author
This manuscript is a good study on induce systemic resistance against plant pathogens.I have some questions and suggestions;
Which kind of induce resistance (SAR or ISR) was activated against this pathoge
Please add the novelity of this study.
The section of discussion needs to improve.
Add the importance of the main enzymes involve induce resistance in introduction section.
Round 2
Reviewer 1 Report
There are some English and/or other mistakes in the text. The authors should consider these lines of text: 23, 32(spp. not italic), 36, 39, 48, 52, 79, 83, 95 (RBOHD?), 108, 114, 124, 146 (the authors should elaborate this. It is said that seedlings were transferred together with the MS medium to the dishes. This is crucial for the success of the experiment. Roots should touch the fungal mycelium. How it is transferred with the MS medium, all medium, or some agar parts with seedlings? For obtaining meaningful results an encounter of flax seedlings with the fungus is necessary. Photographs may help.), 152, 179, 188, 192, 207, 210 (MgCl2),211, 229 (Fusarium culmorum were grown on PDA for 48 h may not be enough time for assessing ABA. For meaningful results, more time might be needed. Two days of growth may not be enough. Maybe 7-10 days will be more meaningful. I suggest the authors should redo the experiment stated in '2.9 Treatment of F. culmorum with ABA' with the dishes 7-10 days of fungal growth and discuss the findings in detail in the manuscript.), 237, 255, 332-333, 340, 360, 364, 367, 384, 391, 398, 405, 407, 409, 410, 418, 420, 425,427, 430, 434, 439, 449, 456, 457, 462, 469, 487, 497, References section also should be checked. The authors should check the lines especially 519, 522, 525, 532, 538, 539, 555, 566, 569, 570, 572, 573, 580, 581, 613, 614, 622, 623, 627
Author Response
Reviewer 1
There are some English and/or other mistakes in the text. The authors should consider these lines of text: 23, 32(spp. not italic), 36, 39, 48, 52, 79, 83, 95 (RBOHD?), 108, 114, 124
> We corrected typos and mistakes throughout the text. Also, we used RboH instead of NADPox, as this is more frequently used abbreviation of the gene. Additionally, we enhanced the Abbreviation section.
146 (the authors should elaborate this. It is said that seedlings were transferred together with the MS medium to the dishes. This is crucial for the success of the experiment. Roots should touch the fungal mycelium. How it is transferred with the MS medium, all medium, or some agar parts with seedlings? For obtaining meaningful results an encounter of flax seedlings with the fungus is necessary. Photographs may help.),
> We corrected the description of the experiment and added photos as supplementary material.
152, 179, 188, 192, 207, 210 (MgCl2),211, 229 (Fusarium culmorum were grown on PDA for 48 h may not be enough time for assessing ABA. For meaningful results, more time might be needed. Two days of growth may not be enough. Maybe 7-10 days will be more meaningful. I suggest the authors should redo the experiment stated in '2.9 Treatment of F. culmorum with ABA' with the dishes 7-10 days of fungal growth and discuss the findings in detail in the manuscript.),
> The time of 48 hours were written there by mistake. The experiment lasted for 14 days actually, and the fungus growth was monitored daily. The mistake was corrected.
237, 255, 332-333, 340, 360, 364, 367, 384, 391, 398, 405, 407, 409, 410, 418, 420, 425,427, 430, 434, 439, 449, 456, 457, 462, 469, 487, 497,
> We went through the text of the manuscript and amended all editorial and language mistakes/errors
References section also should be checked. The authors should check the lines especially 519, 522, 525, 532, 538, 539, 555, 566, 569, 570, 572, 573, 580, 581, 613, 614, 622, 623, 627
> We checked the References section and added missing information. We also noticed, that in some cases changes to the display of the text appeared, depending on the version of text editor we were using, with pdf version being the safest.